# High Carbonyl Graphene Oxide Suppresses Colorectal Cancer Cell Proliferation and Migration by Inducing Ferroptosis via the System Xc−/GSH/GPX4 Axis

**DOI:** 10.3390/pharmaceutics16121605

**Published:** 2024-12-17

**Authors:** Xiecheng Zhou, Qixing Zhang, Haoran Zhu, Guangxiong Ouyang, Xin Wang, Yuankun Cai

**Affiliations:** 1Department of General Surgery, The Fifth People’s Hospital of Shanghai, Fudan University, Shanghai 200240, China; isxiechengzhou@163.com (X.Z.); haoran0975@163.com (H.Z.);; 2Center of Community-Based Health Research, Fudan University, Shanghai 200240, China; zqxdoctor@163.com; 3Department of Pediatrics, The Fifth People’s Hospital of Shanghai, Fudan University, Shanghai 200240, China

**Keywords:** colorectal cancer, graphene oxide, ferroptosis, reactive oxygen species

## Abstract

Background/Objectives: Colorectal cancer (CRC) is characterized by a high rate of both incidence and mortality, and its treatment outcomes are often affected by recurrence and drug resistance. Ferroptosis, an iron-dependent programmed cell death mechanism triggered by lipid peroxidation, has recently gained attention as a potential therapeutic target. Graphene oxide (GO), known for its oxygen-containing functional groups, biocompatibility, and potential for functionalization, holds promise in cancer treatment. However, its role in ferroptosis induction in CRC remains underexplored. The objective of this study was to investigate the effects of High Carbonyl Graphene Oxide (HC-GO) on ferroptosis in CRC and elucidate the underlying mechanisms. Methods: In vitro assays were conducted to evaluate the impact of HC-GO on CRC cell proliferation, mitochondrial function, iron accumulation, lipid peroxidation, and reactive oxygen species (ROS) production. The ferroptosis inhibitor Fer-1 was used to confirm the role of ferroptosis in HC-GO’s anti-tumor effects. In vivo, the anti-tumor activity of HC-GO was assessed in a CRC xenograft model, with organ toxicity evaluated. Results: HC-GO significantly inhibited CRC cell proliferation, induced mitochondrial damage, and enhanced iron accumulation, lipid peroxidation, and ROS production. It also downregulated the ferroptosis-inhibiting proteins GPX4 and SLC7A11, which were reversed by Fer-1, confirming the involvement of ferroptosis in HC-GO’s anti-cancer effects. In vivo, HC-GO significantly suppressed tumor growth without noticeable toxicity to vital organs. Conclusions: HC-GO triggered ferroptosis in CRC cells by suppressing the System Xc−/GSH/GPX4 pathway, providing a novel therapeutic strategy for CRC treatment. These findings suggest HC-GO as a promising nanomedicine for clinical application, warranting further investigation to explore its potential in CRC therapy.

## 1. Introduction

Colorectal cancer (CRC) is now a significant global health concern. As reported in the 2020 Global Cancer Statistics, CRC is the third most common cancer by incidence and the second leading cause of cancer-related death [1,2]. Despite significant advances in CRC treatment through surgery, chemotherapy, radiotherapy, targeted therapy, and immunotherapy [3,4,5], patients continue to encounter issues like cancer recurrence, heightened drug resistance, and treatment-induced toxicity [6,7,8]. Therefore, there is an urgent need to develop new CRC treatment strategies.

Ferroptosis is a novel form of programmed cell death, distinct from apoptosis, and is dependent on the formation and accumulation of lipid radicals driven by iron [9]. The key characteristic of ferroptosis is the overproduction of lipid reactive oxygen species (ROS) inside cells, which leads to fatal lipid peroxidation when the cell’s antioxidant capacity is diminished. Therefore, the sensitivity of ferroptosis is tightly regulated by biological processes that maintain redox homeostasis, including ROS production and the biosynthesis of glutathione (GSH) [9,10]. Glutathione is a key intracellular antioxidant, and glutathione peroxidase 4 (GPX4) uses GSH to reduce lipid peroxides, converting GSH to oxidized glutathione (GSSG), thereby preventing membrane lipid peroxidation and inhibiting ferroptosis [11]. Inducing ferroptosis not only inhibits tumor growth but also improves the effectiveness of immunotherapy and could potentially overcome resistance to current cancer treatments [12,13]. For instance, Huang Y et al. showed that the inhibition of nuclear factor erythroid 2-related factor 2 (Nrf2) increases CRC chemotherapy sensitivity by promoting ferroptosis and apoptosis [14]. Wen RJ et al. discovered that Baicalin triggers ferroptosis in osteosarcoma (OS) by increasing iron accumulation and ROS production while identifying a novel Nrf2/xCT (SLC7A11)/GPX4 regulatory axis involved in Baicalin-induced ferroptosis in OS [15]. Furthermore, a recent study demonstrated that cyclin-dependent kinase 1 (CDK1) suppresses ferroptosis, contributing to oxaliplatin resistance. This suggests that CDK1 inhibitors could offer a promising strategy for treating oxaliplatin-resistant CRC patients [16]. In summary, therapies that induce ferroptosis offer a promising new avenue for overcoming resistance to conventional CRC treatments.

Nanotechnology is an ever-evolving field, and its potential biomedical applications in nanomedicine are continually emerging, especially in cancer treatment. GO, as a functional derivative of graphene, has water solubility, a high surface area, excellent biocompatibility, and strong near-infrared (NIR) absorption, indicating promising applications, particularly in anticancer drug delivery, controlled release, and reducing side effects [17,18]. Studies had indicated that GO exhibits strong inhibitory effects against a range of cancers, including lung cancer [19], breast cancer [20], cervical cancer [21], prostate cancer [22], and osteosarcoma [23], suggesting its practical application value in the field of anti-tumor therapy. Notably, numerous studies had highlighted that the decrease in cell viability induced by GO is mainly due to an increase in intracellular ROS. The accumulation of ROS can result in mitochondrial dysfunction and DNA damage, ultimately reducing cancer cell viability [24,25]. For instance, in phototherapy, GO-based nanocomposites can produce ROS when exposed to near-infrared light, which induces photothermal and photodynamic effects, effectively suppressing tumor growth [26,27]. Some researchers had found that GO nanosheets hinder the proliferation and migration of prostate cancer cells (PC-3) and human neuroblastoma cells (SH-SY5Y) by elevating intracellular ROS levels and inhibiting prostaglandin-mediated inflammatory responses [28]. Liu X et al. developed graphene oxide-grafted magnetic nanorings for magnetothermal therapy, demonstrating enhanced antitumor efficacy closely related to ROS-associated immune responses [29]. Additionally, Shen J et al. reported that GO exerts antitumor effects in CRC through ROS-dependent autophagy and apoptosis, involving the AMPK/mTOR/ULK-1 pathway [30]. Overall, the anti-cancer effects of GO are closely related to ROS production. It is worth noting that the Hummer method is the most commonly used strategy for synthesizing GO [31]. Although different synthesis methods produce GO with variations in characteristics like lateral size and the number of oxygen-containing groups, their fundamental properties are similar. The water solubility of GO is due to the presence of oxygen-containing groups such as hydroxyl, carbonyl, and epoxy groups. Among these, the carbonyl group plays a key role and forms the basis for the functionalization of GO [32]. Research indicates that carbonyl groups act as active sites in redox reactions, effectively engaging in electron transfer, which enhances the redox activity of the material [33]. Researchers had found that the removal of carbonyl, carboxyl, or hydroxyl groups from graphene quantum dots (GQDs) decreases their ROS generation activity, with the removal of carbonyl groups resulting in the lowest ROS production. This suggests that oxygen-containing groups, particularly the carbonyl group, play a crucial role in ROS generation [34]. Moreover, existing studies indicate that the anti-tumor effects of GO are closely related to the generation of intracellular ROS, while one prominent feature of ferroptosis is the excessive production of intracellular lipid ROS. Currently, there is no literature exploring the relationship between GO and its derivatives with ferroptosis. Therefore, this study aims to enhance the carbonyl oxygen content of GO to synthesize High Carbonyl Graphene Oxide (HC-GO) and investigate its relationship with ferroptosis in CRC, as well as the underlying mechanisms. This study examines the impact of HC-GO on the biological functions of CRC through both in vitro and in vivo trials, focusing on its regulation of CRC via the ferroptosis pathway. This could guide us in identifying new molecular targets for CRC treatment and discovering effective therapeutic drugs, which may improve the efficacy of CRC treatment.

## 2. Materials and Methods

### 2.1. Preparation of GO and HC-GO

The following procedure was carried out to prepare GO. Initially, 4.0 g of graphite powder (500 mesh) was combined with 25.0 mL of 98% concentrated sulfuric acid and stirred for 1 h using a magnetic stirrer. Subsequently, 2.0 g of sodium nitrate (NaNO_3_) was added to the mixture, which was then stirred for an additional hour at 0.0 °C. Following this, 12.0 g of potassium permanganate (KMnO_4_) was gradually added in small portions, ensuring the temperature remained below 5.0 °C throughout the process. The temperature was then increased to 35.0 °C, and the mixture was stirred for another 2 h. Gradually, 180.0 mL of deionized water was introduced, and the mixture was stirred at 95.0 °C for 15 min. Finally, 20.0 mL of 30% hydrogen peroxide (H_2_O_2_) was added dropwise, and the solution was washed three times with 1.0 mol/L dilute hydrochloric acid. The resulting GO was obtained by dialysis for one week using a dialysis bag with a molecular weight cutoff of 8000–12,000 Da.

Previous studies [35,36] have shown that increasing the mesh size of graphite and extending the oxidation reaction time can enhance the content of carbonyl oxygen in GO. Therefore, in this study, 8000-mesh graphite powder was used to prepare HC-GO. After adding 2.0 g of NaNO_3_, the stirring time at 0.0 °C was extended from 1 h to 2 h. In the final step, 50.0 mL of 50% hydrogen peroxide was added, while the other steps remained unchanged.

### 2.2. Characterization of HC-GO

A 1.0 mg/mL solution of GO and HC-GO was drop-cast onto a silicon wafer and dried at 37 °C before conducting Raman spectroscopy and X-ray photoelectron spectroscopy (XPS) analysis. Specifically, Raman analysis was performed using a confocal micro-Raman spectrometer (InVia, Renishaw, UK) with an excitation wavelength of 514.0 nm. The surface morphology of the samples was observed using a field emission scanning electron microscope (FESEM, Ultra Plus, Zeiss, Germany) operated at 20 kV. The elemental composition and chemical states of the samples were determined using an XPS (AXIS UltraDLD, Shimadzu, Japan) with a 1486.6 eV Al Kα excitation source, and the results were calibrated against the C 1s peak at 285.0 eV. The surface of HC-GO was analyzed using Atomic Force Microscopy (AFM) with a Scanning Probe Microscope (SPM, BRUKER Dimension Icon, DE, Billerica, MA, USA).

### 2.3. Cell Culture

HCT116 and HCT15 cells were obtained from the Cell Bank of the Chinese Academy of Sciences. HCT116 cells were cultured in Dulbecco’s Modified Eagle Medium (DMEM, Solarbio, Beijing, China) supplemented with 10% fetal bovine serum (ExCell, Suzhou, China), 100 U/mL penicillin, and 100 μg/mL streptomycin. HCT15 cells were cultured in RPMI-1640 (Solarbio, Beijing, China) medium supplemented with 10% fetal bovine serum, 100 U/mL penicillin, and 100 μg/mL streptomycin.The cells were maintained at 37 °C in a 5% CO_2_ humidified incubator. To investigate the effects of HC-GO on CRC cells, the cells were treated with varying concentrations of HC-GO (0 and 10 μg/mL). The selection of HC-GO concentrations was based on previous studies and preliminary experimental data [21,30].

### 2.4. Cell Proliferation and Migration Assays

#### 2.4.1. Plate Clone Formation Assay

Logarithmically growing cells were seeded into 12-well plates at a density of 5 × 10^2^ cells per well and cultured for 24 h. The cells were then treated with medium containing 0 and 10 μg/mL HC-GO. After 14 days of incubation, visible colonies had formed. The colonies were fixed with precooled methanol at 4 °C for 15 min and then stained with crystal violet dye (ShareBio, Shanghai, China) for 30 min. Finally, the colonies were observed and counted.

#### 2.4.2. CCK-8 Assay

Cells were plated into 96-well plates at a density of 2 × 10^3^ cells per well. Once the cells had adhered, they were treated with medium containing 0 and 10 μg/mL HC-GO for 24, 48, and 72 h. After two washes with serum-free medium, 90 μL of serum-free medium and 10 μL of CCK-8 reagent (ShareBio, Shanghai, China) were added to each well. The cells were incubated at 37 °C for 2 h, and absorbance was measured at 450 nm using a microplate reader (Tecan, Männedorf, Switzerland).

#### 2.4.3. Wound Healing Assays

HCT116 and HCT15 cells were seeded into 12-well plates at a density of 4 × 10^5^ cells per well. Once the cells reached 90% confluence, a scratch was made in the cell monolayer using a 10 μL pipette tip. The cells were then washed twice with PBS and incubated with serum-free medium containing 0 and 10 μg/mL HC-GO. Images of the scratch were captured under a microscope (Leica, GRE, St. Gallen, Switzerland) at 0 and 24 h.

#### 2.4.4. Transwell Migration Assay

Transwell chambers (Corning Inc., Corning, NY, USA) were placed in 24-well plates. A suspension of 2 × 10^5^ cells in 200 μL of serum-free medium containing 0 and 10 μg/mL HC-GO was added to the upper chamber, while 700 μL of complete medium containing serum was added to the lower chamber. The cells were incubated at 37 °C for 24 h. After incubation, the medium in the upper chamber was removed, and the chamber was gently washed twice with PBS. Non-migrated cells in the upper chamber were wiped off, and the migrated cells in the lower chamber were fixed with precooled methanol at 4 °C for 20 min. The cells were then stained with crystal violet dye (ShareBio, Shanghai, China) for 2 h, followed by observation and counting under a microscope (Leica, GRE).

### 2.5. Ferroptosis-Related Assay Measurements

#### 2.5.1. Fe^2+^ Content Assay

Intracellular Fe^2+^ levels were measured using a ferrous ion detection kit (Solarbio, Beijing, China) following the manufacturer’s instructions. After treatment with medium containing 0 and 10 μg/mL HC-GO for 24 h, the cells were lysed by ultrasonic disruption on ice. The absorbance was then measured at 593 nm using a microplate reader (Tecan, Männedorf, Switzerland).

#### 2.5.2. ROS Assay

The levels of ROS were measured using a ROS detection kit (Solarbio, Beijing, China), based on the oxidation of 2′,7′-dichlorodihydrofluorescein diacetate (DCFH-DA). HCT116 and HCT15 cells were seeded into 6-well plates and incubated with medium containing 0 and 10 μg/mL HC-GO for 24 h. The cells were then treated with 10 μmol/L DCFH-DA at room temperature for 20 min. After digestion with 0.25% trypsin-EDTA (Yeasen, Shanghai, China), the cells were collected by centrifugation. They were then resuspended and treated with an additional 10 μM DCFH-DA for another 20 min before being analyzed by flow cytometry.

#### 2.5.3. Lipid ROS Assay

HCT116 and HCT15 cells were cultured with medium containing 0 or 10 μg/mL HC-GO for 24 h. After removing the medium, the cells were carefully washed twice with PBS. Then, 1 mL of fresh complete medium containing 10 μM C11 BODIPY fluorescent probe was added, and the cells were incubated in the dark at 37 °C with 5% CO_2_ for 30 min. Afterward, the cells were washed twice with PBS, digested with 0.25% trypsin-EDTA, and resuspended in 500 μL of PBS. Finally, the cells were analyzed by flow cytometry.

#### 2.5.4. GSH Detection

HCT116 and HCT15 cells were treated with culture medium containing 0 and 10 μg/mL HC-GO for 24 h. After treatment, the medium was removed, and the cells were washed twice with precooled PBS. The cell pellets were collected and resuspended in 0.5 mL of PBS. The cells were then disrupted by ultrasonication, and 0.1 mL of the resulting cell suspension was transferred to a new tube. To this, 0.1 mL of Reagent One was added and mixed thoroughly. The mixture was centrifuged at 3500 rpm for 10 min, and the supernatant was collected for further analysis. The GSH detection working solution was prepared according to the instructions in the GSH Assay Kit (Jian Cheng Technology, Nanjing, China). After thorough mixing by vortexing, absorbance was measured at 405 nm using a microplate reader. A standard curve was generated based on the concentration and absorbance of the standard wells, and the GSH content in the samples was calculated accordingly. The final GSH content is expressed as micromoles of GSH per gram of protein (umol/g protein).

### 2.6. Western Blotting

Cells were collected from culture dishes into Eppendorf tubes, and the supernatant was removed. The cells were resuspended in 1 mL PBS and centrifuged again to remove the supernatant. Based on the cell volume in the Eppendorf tube, 200 μL of protein lysis buffer, 2 μL of 100× protease inhibitor cocktail, and PMSF reagent were added. The mixture was gently mixed and incubated at 4 °C for 2 h for cell lysis. Afterward, the cell lysates were centrifuged at 12,000 rpm for 10 min at 4 °C, and the supernatant was collected. Protein concentration was measured using a BCA protein assay kit (Beyotime, Shanghai, China). Protein samples were then transferred to a polyvinylidene difluoride (PVDF) membrane (Millipore Sigma, St. Louis, MA, USA) via gel electrophoresis. The membrane was blocked with protein-free blocking buffer (ShareBio, Shanghai, China) at room temperature for 15 min. Next, the membrane was incubated overnight at 4 °C with specific primary antibodies against GPX4 (1:500, Zenbio, Chengdu, China), SLC7A11 (1:1000, Abclonal, Wuhan, China), FTH1 (1:1000, Abclonal, Wuhan, China), NANOG (1:3000, Abclonal, Wuhan, China), OCT4 (1:1000, Abclonal, Wuhan, China), CD133 (1:1000, Abclonal, Wuhan, China), and GAPDH (1:50000, Abclonal, Wuhan, China). After washing three times with Tris-buffered saline with 0.1% Tween 20 (TBST), the membrane was incubated with goat anti-rabbit secondary antibody (1:10,000, CST, Danvers, MA, USA) at room temperature for 2 h. The membrane was washed three times with TBST and developed using an ECL detection kit (Epizyme, Shanghai, China). Data were analyzed using ImageJ 1.53 software.

### 2.7. Transmission Electron Microscopy

Cell samples were fixed overnight at 4 °C in 1 mL of 2.5% glutaraldehyde solution then washed four times with PBS, each for 15 min. The samples were further fixed in 1% osmium tetroxide solution (Ted Pella Inc., Redding, CA, USA) at 4 °C for 2 h, followed by four washes with PBS, each lasting 15 min. The samples were then dehydrated through a graded ethanol series (50%, 70%, 80%, 90%), with each step lasting 15 min, and left in 70% ethanol overnight. Next, the samples were dehydrated twice in 100% ethanol for 20 min each. The samples underwent two acetone (Sinopharm, CHN, Shanghai, China) exchanges, each for 15 min. For infiltration, the samples were first treated with a 2:1 mixture of acetone and EMBed 812 embedding medium (SPI, Washington, DC, USA) for 1 h, followed by a 1:2 mixture for 4 h, and finally infiltrated with pure EMBed 812 twice, each for 12 h. The samples were embedded in molds with pure embedding medium and polymerized at 65 °C for at least 48 h. After polymerization, the samples were trimmed into trapezoidal shapes with dimensions smaller than 0.2 mm × 0.2 mm. Ultra-thin sections of 70 nm were cut and stained sequentially with uranyl acetate and lead citrate for 10 min each. The samples were then observed, and images were collected using a transmission electron microscope (FEI, Hillsboro, OR, USA).

### 2.8. Immunohistochemical Analysis

Adherent cells were washed three times with PBS and fixed with precooled methanol at 4 °C for 15 min. After removing the methanol with PBS, the cells were permeabilized with 0.5% Triton X-100 solution at room temperature for 5 min. The cells were then blocked with 1% bovine serum albumin (BSA) (Beyotime, Shanghai, China) at room temperature for 2 h and incubated overnight at 4 °C with primary antibodies against GPX4 (1:50, Zenbio, Chengdu, China) or SLC7A11 (1:100, Abclonal, Wuhan, China). Following three washes with PBST, the cells were incubated with goat anti-rabbit secondary antibody (1:400, CST, USA) at room temperature for 2 h in the dark. The cells were subsequently stained with DAPI for 5 min to visualize the nuclei. Finally, fluorescent images were captured using a fluorescence microscope (Olympus, Tokyo, Japan).

### 2.9. Immunofluorescence Staining

Cover slips were immersed in 75% ethanol for 10 min, dried, and placed into 12-well plates. Digested HCT116 cells (1 × 10^6^ cells/well) were evenly distributed into the wells. After the cells adhered, they were treated with medium containing 0 and 10 μg/mL HC-GO for 24 h. The medium was removed, and the cells were washed three times with PBS. Next, 1 mL of 4% paraformaldehyde was added to each well, and the cells were fixed at room temperature for 15 min, followed by three washes with PBS. The cells were then permeabilized with 1 mL of 0.1% Triton X-100 at room temperature for 30 min and blocked with 5% BSA for 1 h. Each cover slip was incubated with 100 μL of primary antibody against GPX4 (1:100, Zenbio, Chengdu, China) or SLC7A11 (1:200, Abclonal, Wuhan, China) in a humidified chamber at 4 °C overnight. After removing the primary antibody and washing three times with PBS, the cells were incubated with goat anti-rabbit secondary antibody (1:500, CST, USA) at room temperature for 1 h in the dark. After washing three times with PBS, the cells were stained with DAPI (1:1000 dilution) at room temperature in the dark for 10 min. Finally, the cover slips were mounted with glycerol, and fluorescent images were captured and stored using a fluorescence microscope (Olympus, Tokyo, Japan).

### 2.10. Animal Experiment

Fifteen male BALB/C nude mice, aged 4 weeks and weighing 16–17 g, were obtained from Shanghai Chengxi Biotechnology Co., Ltd. (Shanghai, China) and housed under specific pathogen-free (SPF) conditions at the Animal Experiment Center of East China Normal University (Shanghai, China). The mice were maintained on a 12 h light/dark cycle at a temperature range of 20–26 °C. After a 3-day acclimatization period, mice in the experimental group were injected with 2 × 10^6^ HCT116 cells suspended in 0.2 mL of PBS containing 10 μg/mL HC-GO, while control group mice received an equivalent volume of PBS. The injection was administered at the right forelimb near the shoulder region. Fourteen days following the injection, the tumor-bearing mice were euthanized via cervical dislocation, and the tumors were excised and weighed. Tumor size and weight were recorded, and the heart, liver, spleen, lungs, and kidneys were collected for further analysis. Tumor volume was calculated using the formula: volume = (length^2^ × width × π)/6. This study was approved by the Animal Ethics Committee of Fudan University (NO.202404016S).

### 2.11. Histological Analysis

Following euthanasia, the tumors, along with the heart, liver, spleen, lungs, and kidneys, were carefully excised and fixed in 4% formaldehyde. The tissues were subsequently embedded in paraffin, and sections approximately 4 μm thick were prepared. These sections were then stained with hematoxylin and eosin (H&E) for histological examination of the tissue morphology.

### 2.12. Statistical Analysis

Statistical analysis was conducted using GraphPad 9.0 (GraphPad, San Diego, CA, USA). To compare data between two groups, independent sample t-tests were used, while one-way ANOVA was applied for comparisons among multiple groups. The results are presented as mean ± standard deviation, and a *p*-value of less than 0.05 was considered statistically significant.

## 3. Results

### 3.1. Structure and Morphology of HC-GO

We observed the GO we prepared using a scanning electron microscope (ZEISS, Jena, Germany). The results showed that the HC-GO we obtained presented a two-dimensional sheet-like structure with obvious wrinkles and folds on its surface. These wrinkles and folds are beneficial for increasing the material’s specific surface area (Figure 1A). The AFM analysis of HC-GO revealed that it is a monolayer with an arithmetic average roughness (Ra) of 3.28 nm, a root mean square roughness (Rq) of 10.7 nm, a diameter of approximately 3–6 μm, and a height of about 2–3 nm (Figure 1B). Next, we analyzed the prepared HC-GO using a Raman spectrometer and conducted XPS analysis on both GO and HC-GO to identify their characteristics. The Raman spectra indicated [37] a D peak at 1361 cm^−1^, which does not exist in pristine graphite due to crystal symmetry. The intensity of this signal indicates the extent of disruption in the GO lattice and the influence of edge structures. The G peak at 1601 cm^−1^ corresponds to the internal stretching vibrations of sp2 hybridized carbon atoms, and its intensity reflects the degree of GO lattice arrangement. The ratio of the D peak to the G peak (D/G) was 0.75 (Figure 1C). XPS results [38] revealed that the surfaces of the prepared regular GO and HC-GO were rich in oxygen-containing functional groups. The carbonyl oxygen content in HC-GO was significantly higher than that in regular GO (Figure 1D). Furthermore, we dissolved the prepared HC-GO in DMEM and 1640 culture media at a concentration of 10 μg/mL. The results indicated that HC-GO could dissolve stably in the media without significant aggregation, and no contamination of the culture media was observed after 72 h. In the subsequent experiments, unless otherwise specified, the concentration of HC-GO used in the HC-GO group was 10 μg/mL. In summary, the HC-GO we prepared is suitable for biological experiments and meets the requirements of our study.

### 3.2. HC-GO Inhibits Proliferation and Migration of CRC Cells In Vitro

The plate colony formation assay results showed that the number of colonies formed by HCT116 and HCT15 cells significantly decreased after treatment with 10 μg/mL HC-GO compared with the control group (Figure 2A,B). The CCK-8 assay further confirmed that the proliferation of HCT116 and HCT15 cells was markedly inhibited by HC-GO treatment (Figure 2C,D). Collectively, these findings indicate that HC-GO significantly suppresses the proliferative capacity of CRC cells (HCT116, HCT15). Additionally, the scratch wound healing assay revealed that the rate of wound closure in cells from the control group was notably faster compared with those treated with HC-GO (Figure 2E,F), indicating that HC-GO significantly hinders the migration speed and capability of HCT116 and HCT15 cells. Consistent with these findings, the Transwell migration assay further confirmed that the migratory potential of cells in the HC-GO-treated group was substantially reduced relative to the control group (Figure 2G,H), suggesting that HC-GO has a strong inhibitory effect on the migration capacity of CRC cells. Moreover, the expression analysis of stemness proteins NANOG, OCT4, and CD133 revealed that HC-GO also inhibits the stemness of CRC cells. In conclusion, HC-GO significantly inhibits the proliferation and migration of CRC cells in vitro.

### 3.3. HC-GO Induces Ferroptosis in CRC Cells In Vitro

We investigated the role of HC-GO in regulating ferroptosis in CRC cells. Initially, we analyzed intracellular Fe^2+^ levels, and the results showed that HC-GO treatment significantly increased Fe^2+^ levels in HCT116 and HCT15 cells (Figure 3A). HC-GO also promoted the accumulation of ROS (Figure 3B,C) and lipid ROS (Figure 3D,E) within HCT116 and HCT15 cells. Furthermore, the results indicated that HC-GO treatment markedly reduced GSH levels in both HCT116 and HCT15 cells (Figure 3F). Additionally, HC-GO inhibited the expression of SLC7A11, GPX4, and FTH1 in HCT116 cells (Figure 3G,H). Transmission electron microscopy revealed that HC-GO-treated cells exhibited smaller mitochondria with significantly reduced or absent cristae and an increased mitochondrial membrane density compared with the control group (Figure 3I). These ultrastructural changes are indicative of ferroptosis, suggesting that mitochondria in CRC cells undergo ferroptotic alterations following HC-GO treatment. In summary, these data suggest that HC-GO can induce ferroptosis in CRC cells in vitro.

### 3.4. HC-GO Inhibits CRC Cells and Induces Ferroptosis In Vivo

To explore the inhibitory effects of HC-GO on CRC cells and its potential to induce ferroptosis in vivo, we developed a cell-line-derived xenograft tumor model in nude mice. This model is straightforward and enables efficient monitoring of treatment outcomes. The findings showed that the tumor volume and weight in the HC-GO-treated group were notably smaller compared with the control group (Figure 4A–D). Furthermore, further histological analysis of the excised tumor tissues using HE staining revealed the presence of black particles within and between tumor cells in the HC-GO group, confirming the accumulation of HC-GO particles. Additionally, there were clear necrotic areas within the tumor tissues, with some cells exhibiting ruptured membranes and incomplete structures, indicative of significant damage and necrosis (Figure 5A). These results implied that HC-GO induced necrosis in CRC cells in vivo. To further investigate the occurrence of ferroptosis, we conducted immunohistochemical staining for GPX4 and SLC7A11 in the subcutaneous tumor tissues of the nude mice. The staining analysis revealed that the expression levels of GPX4 and SLC7A11 were significantly lower in the HC-GO-treated group compared with the control group (Figure 5B,C), which was consistent with the findings obtained in vitro. Therefore, HC-GO is capable of inducing ferroptosis in vivo in the nude mouse model. Moreover, HE staining indicated that 10 μg/ml HC-GO did not cause noticeable damage to the heart, liver, spleen, lung, or kidney tissues of the nude mice (Figure 5D).

### 3.5. HC-GO Inhibits CRC Cells Through Ferroptosis Mediated by the System Xc-/GSH/GPX4 Axis

To investigate the role of the antioxidant system Xc−/GSH/GPX4 axis in HC-GO-induced ferroptosis in CRC cells, we employed the ferroptosis inhibitor Fer-1 to block lipid peroxidation effects. The results showed that Fer-1 partially reversed the inhibitory effect of HC-GO on HCT116 cells (Figure 6A–C). Moreover, Fer-1 inhibited the HC-GO-induced increase in both ROS and lipid ROS levels in HCT116 cells (Figure 6D–G). Additionally, Fer-1 prevented the reduction in GSH levels induced by HC-GO in HCT116 cells (Figure 6H). Western blot analysis further confirmed the impact of Fer-1 on ferroptosis-related protein expression levels, revealing that GPX4 and SLC7A11 expression levels in the HC-GO + Fer-1 group were significantly higher than those in the HC-GO group and were similar to the levels observed in the control group (Figure 6I,J). Immunofluorescence assays additionally demonstrated that the green fluorescence signal of GPX4 and the red fluorescence signal of SLC7A11 were notably stronger in the HC-GO + Fer-1 group compared with the HC-GO group, closely resembling the levels seen in the control group (Figure 6K–N). Collectively, these findings indicate that Fer-1 can block the ferroptotic effects induced by HC-GO and that this process is closely related to the System Xc−/GSH/GPX4 axis, thereby confirming that HC-GO inhibits CRC cells through ferroptosis mediated by the System Xc−/GSH/GPX4 axis.

## 4. Discussion

CRC is among the most prevalent and aggressive malignancies of the digestive tract, often carrying mutations in key genes such as KRAS, NRAS, TP53, and BRAF. This cancer is also characterized by high rates of recurrence and metastasis [2,40,41]. Despite these genetic challenges, recent advances in diagnostic and therapeutic techniques have been made. However, the prognosis for many patients remains poor due to the high heterogeneity and individual variability of CRC [42,43]. Recent studies have indicated that ferroptosis, a novel form of programmed cell death, may represent a promising strategy for CRC treatment [44,45]. In this context, although GO nanomaterials have demonstrated potential antitumor effects, the mechanisms of GO in CRC, particularly its relationship with ferroptosis, remain unclear. In this study, we found that HC-GO can inhibit CRC cells by inducing ferroptosis, providing valuable evidence for the relationship between HC-GO and the ferroptosis mechanism.

Since its formal identification in 2012, ferroptosis has been recognized as a distinct form of programmed cell death, different from autophagy, pyroptosis, necrosis, and apoptosis [9,46]. The core mechanism of ferroptosis involves an imbalance in the cellular redox state, leading to iron overload and the accumulation of lipid ROS, ultimately resulting in cell death [47]. The morphological characteristics of ferroptosis include significant mitochondrial shrinkage, increased membrane density, reduction in or loss of mitochondrial cristae, and rupture of the outer mitochondrial membrane [46]. Consistent with these characteristics, our study showed that HC-GO-treated CRC cells exhibited significant mitochondrial damage, with notable increases in intracellular Fe^2+^, ROS, and lipid ROS levels. Additionally, the antioxidant system plays a pivotal role in regulating ferroptosis, with the System Xc−/GSH/GPX4 axis acting as a critical defense mechanism against this form of cell death. Inhibition of System Xc− reduces GSH synthesis and impairs GPX4 activity, thereby triggering ferroptosis. Our findings indicate that in HC-GO-treated CRC cells, the activity of SLC7A11 and GPX4, as well as GSH levels, were significantly diminished. However, treatment with the antioxidant inhibitor Fer-1 reversed these changes, suggesting that HC-GO induces ferroptosis in CRC cells by inhibiting the System Xc−/GSH/GPX4 axis. The aberrant regulation of key factors involved in ferroptosis and their regulators is closely linked to cancer progression, and targeting the ferroptosis network, particularly GPX4 and SLC7A11, may offer new therapeutic options for cancer and other diseases [48,49]. Previous studies have indicated that GPX4 and SLC7A11 are critical targets in the cellular antioxidant response [50]. Additionally, ferroptosis is closely related to tumor cell resistance, with resistant tumor cells being more sensitive to ferroptosis due to their higher iron consumption compared with normal cells [51]. For example, research has shown that when LGR4-mAb targets the Wnt signaling pathway in combination with chemotherapeutic drugs, it can downregulate the key ferroptosis inhibitor SLC7A11, thereby overcoming acquired resistance in CRC [52]. Similarly, inhibiting the KIF20A/NUAK1/PP1β/GPX4 pathway in CRC cells can induce ferroptosis and overcome oxaliplatin resistance [53]. Furthermore, Wen RJ et al. discovered that baicalin induces ferroptosis in osteosarcoma through a novel Nrf2/xCT/GPX4 regulatory axis [15]. Therefore, targeting molecules such as GPX4 and SLC7A11 to induce ferroptosis may provide a promising strategy to overcome tumor resistance and offer a new avenue for cancer treatment.

As a promising two-dimensional material, GO has shown significant potential in cancer treatment through various mechanisms, including direct cytotoxicity, chemosensitization, drug or gene delivery, and phototherapy [17,18,54,55]. GO’s surface is rich in epoxy, carboxyl, and hydroxyl groups, which facilitate functionalization and provide attachment sites for various molecules such as proteins and DNA [56]. Notably, previous studies have shown that GO can enhance the production of ROS and induce DNA damage, leading to cancer cell death [26,27,28,29,30]. In our study, we found that HC-GO treatment substantially suppressed the growth of CRC cells, with a significant increase in both total ROS and lipid ROS levels compared with the control group, in line with previous reports. Furthermore, existing in vitro and in vivo studies have investigated the potential toxicity of GO nanomaterials in key organs, including the brain, liver, spleen, and heart [54]. For instance, previous studies had shown that intravenous injection of 5 mg/kg of nanoscale graphene oxide (NGO) caused adverse effects on major organs such as the liver, kidneys, spleen, and heart [57]. Similarly, some researchers evaluated the acute toxicity, toxicokinetics, and respiratory/cardiovascular safety of intravenously injected dextran-coated GO nanoplatelets (GNP-Dex) at doses ranging from 1 to 500 mg/kg, finding that the maximum tolerated dose (MTD) was between 50 mg/kg ≤ MTD < 125 mg/kg [58]. Mohamed HRH et al. conducted in vivo experiments on Swiss mice by orally administering graphene oxide nanoparticles (10, 20, and 40 mg/kg), showing that the nanoparticles induced chromosomal, DNA, and histological damage in a dose-dependent manner during acute and subacute treatments [59]. In contrast, our results indicated that subcutaneous injection of GO did not cause significant toxicity to the heart, liver, spleen, lungs, or kidneys in mice. These findings suggest that the biotoxicity of GO cannot be overlooked, and addressing the toxicity and improving the biocompatibility of GO are critical for its effective integration into cancer therapy.

## 5. Limitations

This study has several limitations. Firstly, we were unable to elucidate the specific regulatory mechanisms by which HC-GO inhibits the System Xc−/GSH/GPX4 axis. Secondly, the in vivo animal experimental data are limited, as we did not conduct higher-level validations in CRC metastasis models, orthotopic transplantation models, or gene knockout mouse models. Finally, we did not explore the clinical translational potential of HC-GO in CRC in depth.

## 6. Conclusions

In conclusion, our study demonstrates that HC-GO significantly inhibits CRC cell proliferation both in vitro and in vivo through a ferroptosis mechanism mediated by the System Xc−/GSH/GPX4 axis. This is the first research to explore the interaction between GO or its base materials with the ferroptosis pathway, enriching the knowledge base of ferroptosis-related cancer therapies. The study reveals the potential of HC-GO in combating CRC and provides preliminary insights into its specific mechanisms of action, offering new perspectives for CRC diagnosis and treatment. However, the clinical applicability of GO and its base materials in CRC treatment requires further validation.

## Figures and Tables

**Figure 1 pharmaceutics-16-01605-f001:**
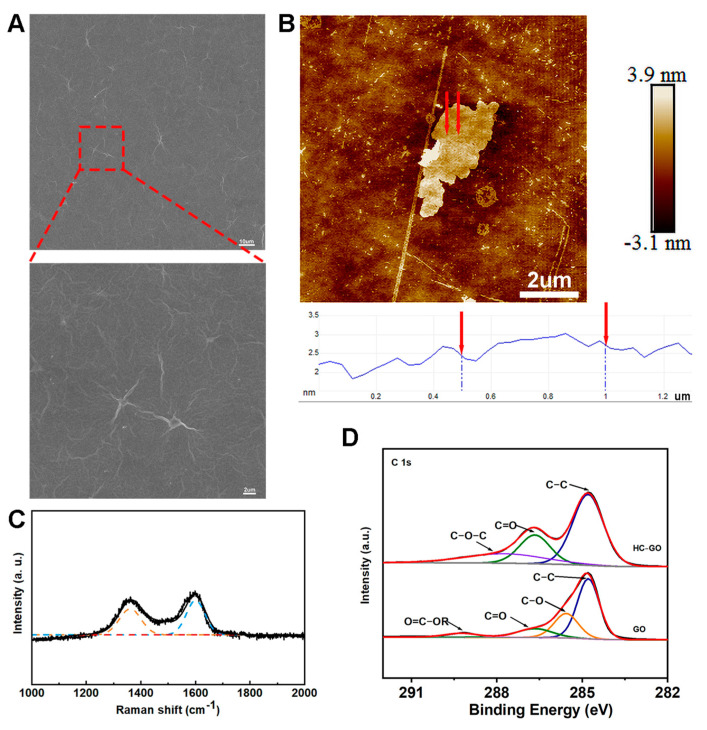
Characterization of HC-GO. (**A**) Scanning electron microscope images of HC-GO, scale bar: 2 μm, 10 μm; (**B**) AFM topography image and the corresponding height distribution graph of HC-GO, scale bar: 2 μm; (**C**) Raman spectra results of HC-GO; (**D**) XPS results comparing regular GO and HC-GO.

**Figure 2 pharmaceutics-16-01605-f002:**
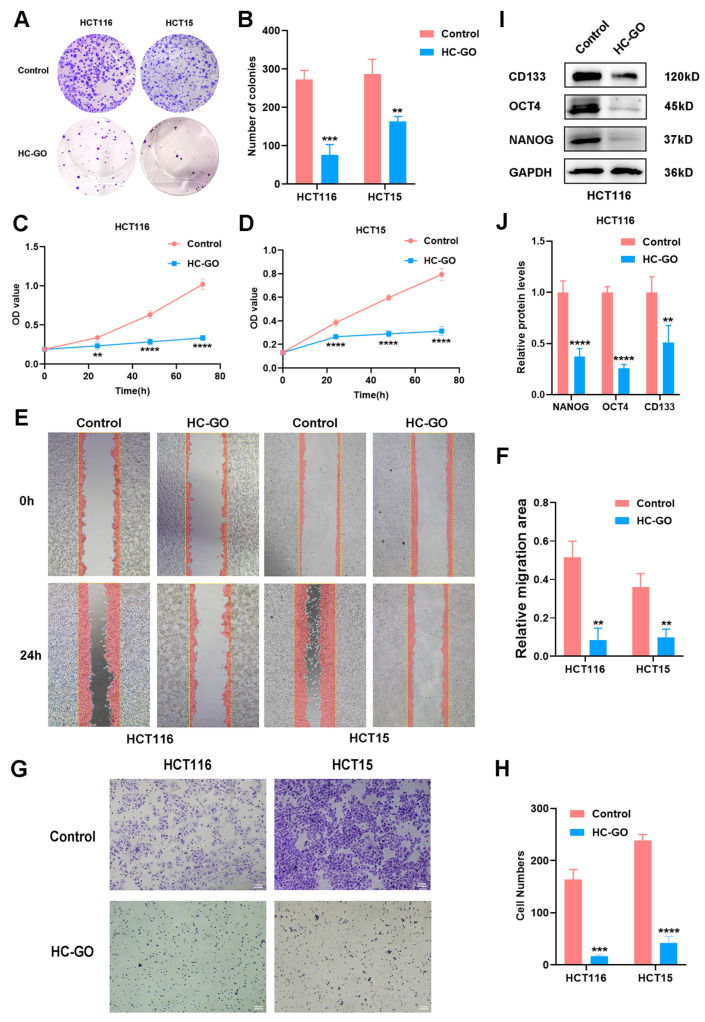
HC-GO significantly inhibited the in vitro proliferation and migration of HCT116 and HCT15 cells. (**A**,**B**) Colony formation assays were used to analyze cell proliferation. (**C**,**D**) CCK-8 assays were conducted to assess cell proliferation. (**E**,**F**) Scratch wound healing assays were employed to analyze cell migration. (**G**,**H**) Transwell assays were performed to measure cell migration capacity. Scale bar: 100 μm. (**I**,**J**) Western blot (WB) analysis was used to assess the expression levels of stemness proteins. Data are presented as mean ± SD. ** *p* < 0.01, *** *p* < 0.001, **** *p* < 0.0001 compared with the control group. All experiments were independently repeated at least three times.

**Figure 3 pharmaceutics-16-01605-f003:**
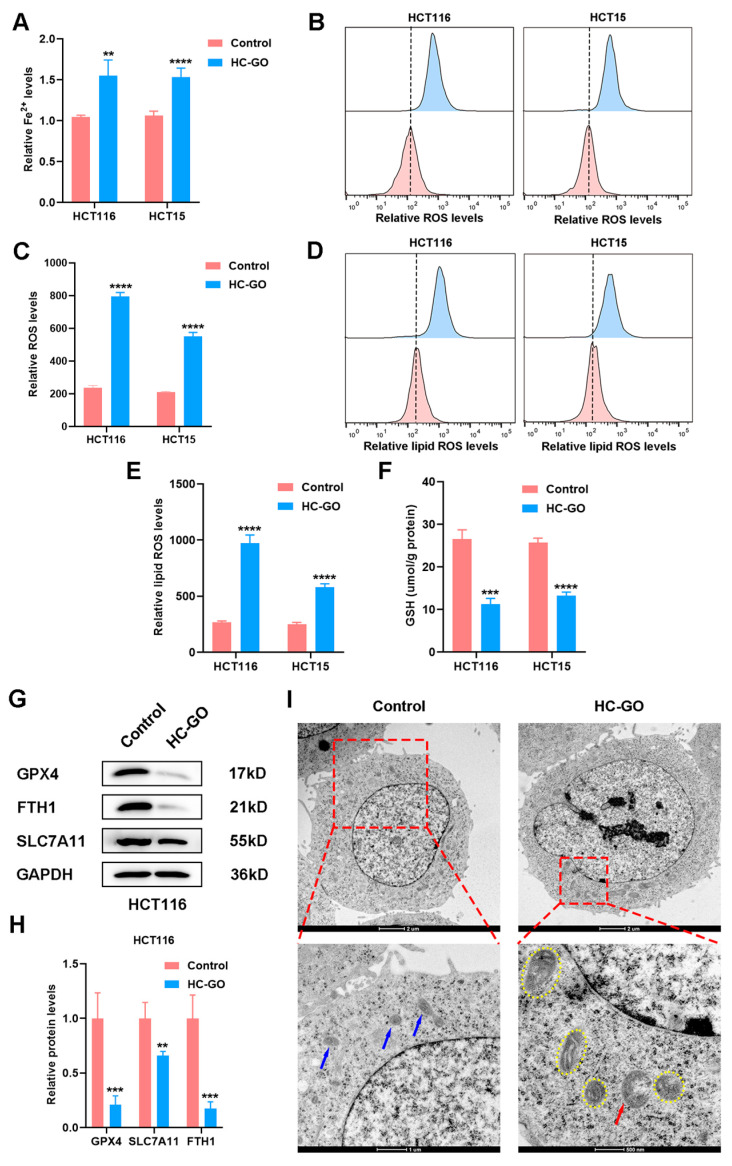
HC-GO induced ferroptosis in HCT116 and HCT15 cells in vitro. (**A**) Analysis of Fe^2+^ levels. (**B**,**C**) Analysis of intracellular ROS levels. (**D**,**E**) Analysis of lipid ROS levels. (**F**) Analysis of GSH levels. (**G**,**H**) Western blot analysis of ferroptosis-related protein expression levels. (**I**) Transmission electron microscopy images of HCT116 cells: blue arrows indicate damaged mitochondria, red arrows indicate mitochondria with vacuolization, and yellow circles highlight structural damage in mitochondria (increased membrane density, reduced cristae, and mitochondrial shrinkage). Scale bars: 1 μm, 2 μm, 500 nm. Data are presented as mean ± SD. ** *p* < 0.01, *** *p* < 0.001, **** *p* < 0.0001 compared with the control group. All experiments were independently repeated at least three times.

**Figure 4 pharmaceutics-16-01605-f004:**
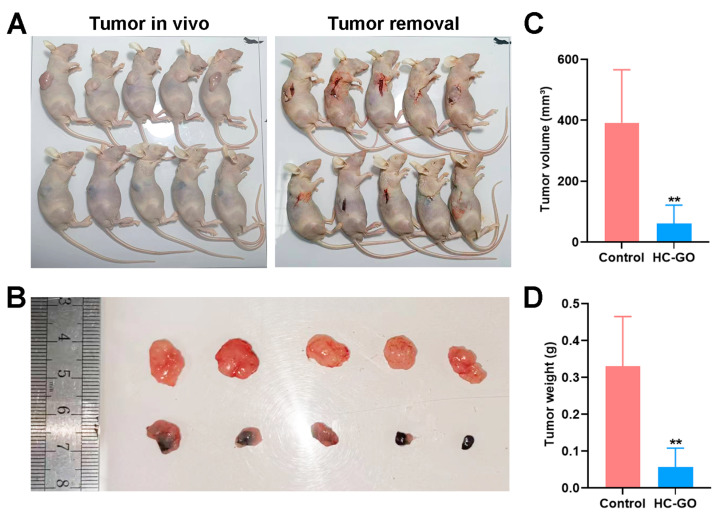
HC-GO inhibited CRC cells in vivo. (**A**,**B**) Tumor appearance; (**C**,**D**) Tumor volume and weight. Data are presented as mean ± SD. ** *p* < 0.01 compared with the control group.

**Figure 5 pharmaceutics-16-01605-f005:**
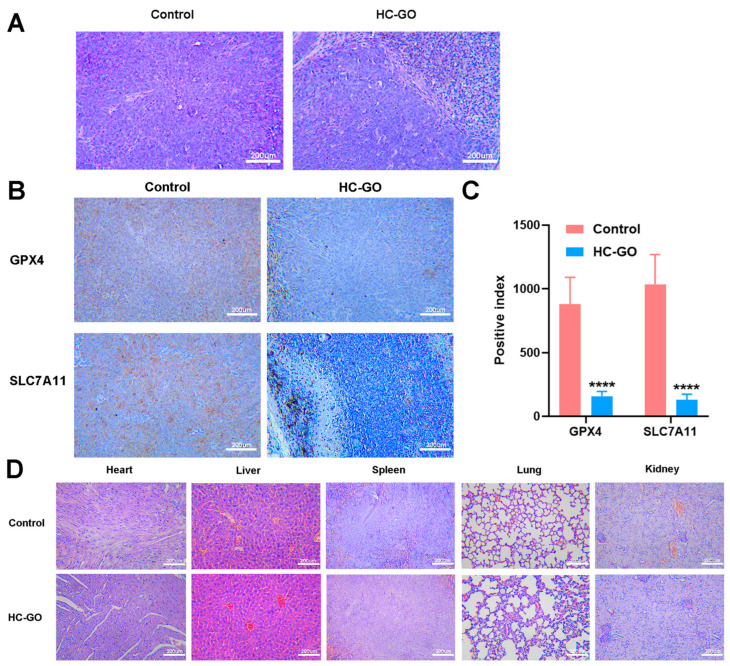
HC-GO induced ferroptosis in vivo. (**A**) HE staining used to assess morphological differences in tumor tissues. Scale bar: 200 μm. (**B**,**C**) Immunohistochemical staining for GPX4 and SLC7A11. Scale bar: 200 μm. (**D**) HE staining used to assess the morphology of mouse heart, liver, spleen, lung, and kidney tissues. Scale bar: 200 μm. Data are presented as mean ± SD. **** *p* < 0.0001compared with the control group.

**Figure 6 pharmaceutics-16-01605-f006:**
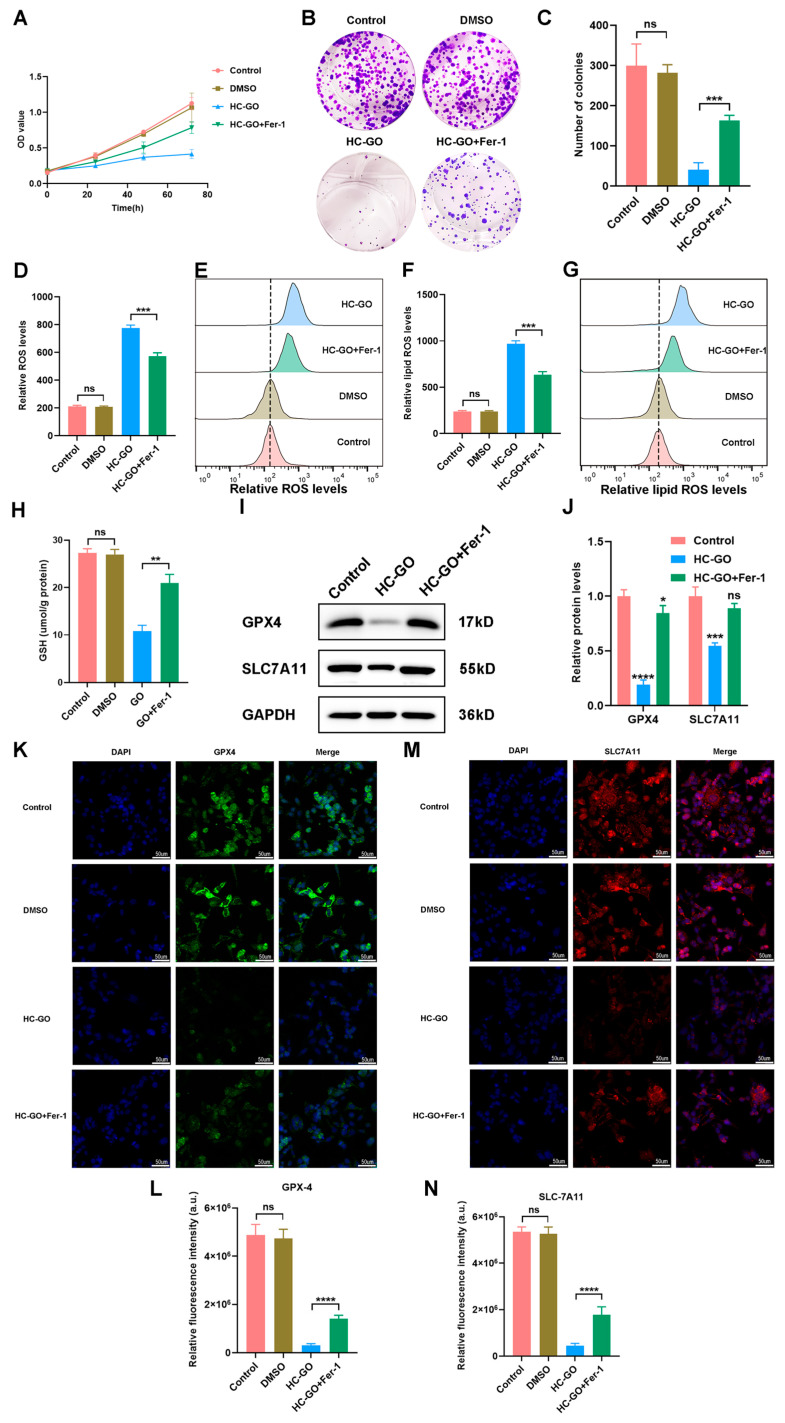
Ferroptosis inhibitor Fer-1 blocked HC-GO-induced ferroptosis in HCT116 cells. The concentration of Fer-1 was 10 μM [39]. (**A**) CCK-8 assay used to analyze cell proliferation; (**B**,**C**) Colony formation assay used to analyze cell proliferation; (**D**,**E**) Fer-1 blocks the increase in intracellular ROS levels induced by HC-GO; (**F**,**G**) Fer-1 blocks the increase in intracellular lipid ROS levels induced by HC-GO; (**H**) Fer-1 blocks the increase in GSH levels induced by HC-GO; (**I**,**J**) Western blot (WB) results show that Fer-1 blocks the decrease in GPX4 and SLC7A11 expression induced by HC-GO; (**K**,**L**) Immunofluorescence shows that Fer-1 blocks the decrease in GPX4 expression induced by HC-GO; (**M**,**N**) Immunofluorescence shows that Fer-1 blocks the decrease in SLC7A11 expression induced by HC-GO. Scale bar: 50 μm. Data are presented as mean ± SD. ^ns^
*p* > 0.05, * *p* < 0.05, ** *p* < 0.01, *** *p* < 0.001, **** *p* < 0.0001compared with the control group/ HC-GO. All experiments were independently repeated at least three times.

## Data Availability

The original contributions presented in this study are included in the article. Further inquiries can be directed to the corresponding author.

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
