# Peer review of "High Carbonyl Graphene Oxide Suppresses Colorectal Cancer Cell Proliferation and Migration by Inducing Ferroptosis via the System Xc−/GSH/GPX4 Axis"

_pharmaceutics, 2024, doi:10.3390/pharmaceutics16121605_

Round 1
Reviewer 1 Report
Comments and Suggestions for Authors
Manuscript ID: pharmaceutics-3330738
Manuscript Title: High Carbonyl Graphene Oxide Suppresses Colorectal Cancer Cell Proliferation and Migration by Inducing Ferroptosis via the System Xc−/GSH/GPX4 Axis.
The research article titled “High Carbonyl Graphene Oxide Suppresses Colorectal Cancer Cell Proliferation and Migration by Inducing Ferroptosis via the System Xc−/GSH/GPX4 Axis” is written in a very scientific way and presents comprehensive details about sensing platforms for HC-GO induces ferroptosis by inhibiting the System Xc−/GSH/GPX4 axis, providing a novel therapeutic avenue for CRC treatment. HC-GO's potential as a nanomedicine for clinical application warrants further investigation, offering a promising new strategy for combating CRC. I recommend publishing this work with the minor revisions suggested below.
Comments:
1. In abstract the formimg of the written patten should be in the same tense. Mixing of tense is not correct.
2. In section 2, the sentences
“2.1. Preparation of GO and HC-GO
First, 4 g of graphite (500 mesh) was mixed with 25 ml of 98% concentrated sulfuric acid and stirred with a magnetic stirrer for 1 hour. Then, 2.0 g of sodium nitrate (NaNO3) was added, and the mixture was stirred for another hour at 0°C. Subsequently, 12.0 g of potassium permanganate (KMnO4) was slowly added in batches, keeping the temperature below 5°C throughout the process. After that, the temperature was raised to 35°C, and stirring continued for 2 more hours. Gradually, 180 ml of deionized water was added, and the mixture was stirred at 95°C for 15 minutes. Finally, 20 ml of 30% hydrogen peroxide (H2O2) was slowly added, followed by three washes with 1 mol/L dilute hydrochloric acid solution. The GO was then obtained by dialyzing the solution for one week using a dialysis bag with a molecular weight cutoff of 8000-12000 Da.” should be corrected as “2.1. Preparation of GO and HC-GO
First, 4.0 g of graphite (500 mesh) was mixed with 25.0 ml of 98% concentrated sulfuric acid and stirred with a magnetic stirrer for 1 hour. Then, 2.0 g of sodium nitrate (NaNO3) was added, and the mixture was stirred for another hour at 0.0°C. Subsequently, 12.0 g of potassium permanganate (KMnO4) was slowly added in batches, keeping the temperature below 5°C throughout the process. After that, the temperature was raised to 35.0°C, and stirring continued for 2 more hours. Gradually, 180 ml of deionized water was added, and the mixture was stirred at 95.0°C for 15 minutes. Finally, 20.0 ml of 30% hydrogen peroxide (H2O2) was slowly added, followed by three washes with 1.0 mol/L dilute hydrochloric acid solution. The GO was then obtained by dialyzing the solution for one week using a dialysis bag with a molecular weight cutoff of 8000-12000 Da.” It is need to do thecorrections of all decimal all over the manuscript.
3. Figure 4 is too long. Make it into 2 figures and explain the results accordilingly.
4. The Figures 5E and G need to be more clear.
Comments on the Quality of English Language
Author Response
Response to Reviewer 1 Comments
|
||
1. Summary |
|
|
We sincerely appreciate the time and effort you have dedicated to reviewing our manuscript and providing valuable feedback. Below, we present detailed responses to your comments, displayed in black text and numbered to guide our revisions. Our responses are provided in red text. Additionally, the corresponding modifications in the manuscript are highlighted in red for clarity.
|
||
2. Questions for General Evaluation |
Reviewer’s Evaluation |
Response and Revisions |
Does the introduction provide sufficient background and include all relevant references? |
Yes |
Thank you for your positive evaluation |
Are all the cited references relevant to the research? |
Yes |
Thank you for your positive evaluation |
Is the research design appropriate? |
Can be improved |
Thank you for your suggestion. We will place more emphasis on the rationality of research design in our future studies. |
Are the methods adequately described? |
Can be improved |
Thank you for your feedback. In future research, we will place greater emphasis on the description of methods. |
Are the results clearly presented? |
Can be improved |
Thank you for your feedback. We will place greater emphasis on the presentation of results in our future research. |
Are the conclusions supported by the results? |
Yes |
Thank you for your positive evaluation |
3. Point-by-point response to Comments and Suggestions for Authors |
||
Comments 1: In abstract the formimg of the written patten should be in the same tense. Mixing of tense is not correct.
|
||
Response 1: First of all, I would like to sincerely thank you for your insightful comments and for taking the time to review our manuscript. We have revised the abstract, and in order to maintain consistency, I chose to use the past tense, as the abstract describes research that has already been conducted. Changes in the text: see Page 1, lines 21-22, 29-32.
|
||
Comments 2: In section 2, the sentences “2.1. Preparation of GO and HC-GO First, 4 g of graphite (500 mesh) was mixed with 25 ml of 98% concentrated sulfuric acid and stirred with a magnetic stirrer for 1 hour. Then, 2.0 g of sodium nitrate (NaNO3) was added, and the mixture was stirred for another hour at 0°C. Subsequently, 12.0 g of potassium permanganate (KMnO4) was slowly added in batches, keeping the temperature below 5°C throughout the process. After that, the temperature was raised to 35°C, and stirring continued for 2 more hours. Gradually, 180 ml of deionized water was added, and the mixture was stirred at 95°C for 15 minutes. Finally, 20 ml of 30% hydrogen peroxide (H2O2) was slowly added, followed by three washes with 1 mol/L dilute hydrochloric acid solution. The GO was then obtained by dialyzing the solution for one week using a dialysis bag with a molecular weight cutoff of 8000-12000 Da.” should be corrected as “2.1. Preparation of GO and HC-GO
First, 4.0 g of graphite (500 mesh) was mixed with 25.0 ml of 98% concentrated sulfuric acid and stirred with a magnetic stirrer for 1 hour. Then, 2.0 g of sodium nitrate (NaNO3) was added, and the mixture was stirred for another hour at 0.0°C. Subsequently, 12.0 g of potassium permanganate (KMnO4) was slowly added in batches, keeping the temperature below 5°C throughout the process. After that, the temperature was raised to 35.0°C, and stirring continued for 2 more hours. Gradually, 180 ml of deionized water was added, and the mixture was stirred at 95.0°C for 15 minutes. Finally, 20.0 ml of 30% hydrogen peroxide (H2O2) was slowly added, followed by three washes with 1.0 mol/L dilute hydrochloric acid solution. The GO was then obtained by dialyzing the solution for one week using a dialysis bag with a molecular weight cutoff of 8000-12000 Da.” It is need to do thecorrections of all decimal all over the manuscript.
Response 2: Thank you for the detailed suggestion. We have carefully revised Section 2.1 and corrected the formatting of all decimal values as suggested. Additionally, we have reviewed the entire manuscript to ensure consistency in the use of decimals throughout. Changes in the text: see Pages 3-4, lines 112-128.
|
||
Comments 3: Figure 4 is too long. Make it into 2 figures and explain the results accordilingly.
|
||
Response 3: Thank you for the suggestion. We have divided Figure 4 into two separate figures to enhance readability and clarity. The corresponding explanations have been updated in the manuscript to align with the revised figures. Changes in the text: see Pages 12-13, lines 391-409.
Comments 4: The Figures 5E and G need to be more clear.
Response 4: Thank you for your comments. We have made efforts to improve the clarity of the original figures 5E and 5G (now Figures 6E and 6G, as we have separated the original Figure 4 into two independent figures, Figures 4 and 5). A clearer version of these figures has been provided in the supplementary file for your review. Please let us know if further adjustments are needed. Changes in the text: see Page 15, Figure 6.
|
- Response to Comments on the Quality of English Language
Point 1: None
Response 1: not applicable.
- Additional clarifications
We have made every effort to improve the manuscript and have implemented several revisions. These adjustments do not alter the overall content or structure of the paper. We have not listed all the changes here; instead, we have highlighted the revised sections in red text within the manuscript. We sincerely appreciate your hard work and hope that our revisions meet your expectations. Once again, we are truly grateful for your insightful comments and suggestions.
Reviewer 2 Report
Comments and Suggestions for Authors
The authors reported that high-carbonyl graphene oxide is a promising material for cancer treatment, supported by thorough materials characterizations, ex vivo, and in vivo demonstrations. The results are significant and potentially impactful to a broad audience. However, there are several points that need to be addressed:
-
Animal Model Choice: The in vivo demonstration focuses on treating a subcutaneous tumor with intratumor injection of the materials. This model has a vague correlation with colorectal cancer applications. The authors should provide a detailed rationale for their choice of animal model and discuss its relevance to colorectal cancer. Additionally, they should consider suggesting future studies using more relevant models.
-
Immunostaining for Ferroptosis: Current immunostaining results in Figure 5 are from HCT116 cells, instead of tissue collected from the animals. The characterization of in vivo treatment lacks immunostaining results to illustrate ferroptosis in the tumor. The authors should include these results to confirm that the tumor treatment effect is due to ferroptosis rather than other potential toxic effects of the chemical.
Author Response
Response to Reviewer 2 Comments
|
||
1. Summary |
|
|
We sincerely appreciate the time and effort you have dedicated to reviewing our manuscript and providing valuable feedback. Below, we present detailed responses to your comments, displayed in black text and numbered to guide our revisions. Our responses are provided in red text. Additionally, the corresponding modifications in the manuscript are highlighted in red for clarity.
|
||
2. Questions for General Evaluation |
Reviewer’s Evaluation |
Response and Revisions |
Does the introduction provide sufficient background and include all relevant references? |
Yes |
Thank you for your positive evaluation |
Are all the cited references relevant to the research? |
Yes |
Thank you for your positive evaluation |
Is the research design appropriate? |
Yes |
Thank you for your positive evaluation |
Are the methods adequately described? |
Yes |
Thank you for your positive evaluation |
Are the results clearly presented? |
Yes |
Thank you for your positive evaluation |
Are the conclusions supported by the results? |
Can be improved |
Thank you for your feedback. We had refined the presentation of the results to enhance clarity and address the suggested improvements. |
3. Point-by-point response to Comments and Suggestions for Authors |
||
Comments 1: Animal Model Choice: The in vivo demonstration focuses on treating a subcutaneous tumor with intratumor injection of the materials. This model has a vague correlation with colorectal cancer applications. The authors should provide a detailed rationale for their choice of animal model and discuss its relevance to colorectal cancer. Additionally, they should consider suggesting future studies using more relevant models.
|
||
Response 1: Thank you for raising this important point. We chose the cell-line-derived xenograft tumor model primarily because of its simplicity and ease of monitoring treatment effects, which allows us to assess the potential therapeutic efficacy of the materials. However, we acknowledge that its relevance to colorectal cancer is limited. To address this, we have included the rationale for selecting this model and discussed its limitations in the revised manuscript. Additionally, we recommend that future studies use orthotopic or patient-derived xenograft models to better simulate the colorectal cancer microenvironment and enhance the clinical relevance of the findings. Changes in the text: see Pages 12, 17; lines 382-384, 512-515.
|
||
Comments 2: Immunostaining for Ferroptosis: Current immunostaining results in Figure 5 are from HCT116 cells, instead of tissue collected from the animals. The characterization of in vivo treatment lacks immunostaining results to illustrate ferroptosis in the tumor. The authors should include these results to confirm that the tumor treatment effect is due to ferroptosis rather than other potential toxic effects of the chemical. |
||
|
||
Response 2: Thank you for your suggestion. We acknowledge the importance of confirming ferroptosis in in vivo tumor tissues. Therefore, we performed immunohistochemical staining for GPX4 and SLC7A11 on the xenograft tumors generated in mice (originally Figure 4F, now Figure 5B) to demonstrate ferroptosis in the tumors. We appreciate your recommendation to further demonstrate ferroptosis in the xenograft tumors using immunofluorescence staining, and we believe this is a valuable approach. We will consider this in our future research. Changes in the text: see Page 13, Figure 5B.
|
- Response to Comments on the Quality of English Language
Point 1: None
Response 1: Not applicable.
- Additional clarifications
We have made every effort to improve the manuscript and have implemented several revisions. These adjustments do not alter the overall content or structure of the paper. We have not listed all the changes here; instead, we have highlighted the revised sections in red text within the manuscript. We sincerely appreciate your hard work and hope that our revisions meet your expectations. Once again, we are truly grateful for your insightful comments and suggestions.
Reviewer 3 Report
Comments and Suggestions for Authors
This study revealed that HC-GO could inhibit cell proliferation by inducing ferroptosis in colorectal cancer cells. The authors also demonstrated the effect of HC-GO in a xenograft mice model and proposed the intracellular pathway that induces ferroptosis. This manuscript presents extensive experimental results on the effects of HC-GO on the ferroptosis pathway through in vitro and in vivo studies, and the manuscript is well organized. Here are some considerations:
- It seems a bit awkward to overuse the first author name of each reference in the introduction or discussion section. It is sufficient to describe only the main content and meaning of the reference.
- What is Xc−? Is it different from xCT? Please explain.
- In the flow cytometry results for ROS levels, the notation Relative ROS levels is written on the vertical axis, which gives the illusion that the height of the plot represents the ROS level. As far as I understand, the ROS level corresponds to the position on the horizontal axis, so this needs to be distinguished. Also, please provide information on cell counts so that we can know the number of cells corresponding to each plot in the flow cytometry.
- Images of nude mice in tumor removal status do not need to be presented in the main figure.
Author Response
Response to Reviewer 3 Comments
|
||
1. Summary |
|
|
We sincerely appreciate the time and effort you have dedicated to reviewing our manuscript and providing valuable feedback. Below, we present detailed responses to your comments, displayed in black text and numbered to guide our revisions. Our responses are provided in red text. Additionally, the corresponding modifications in the manuscript are highlighted in red for clarity.
|
||
2. Questions for General Evaluation |
Reviewer’s Evaluation |
Response and Revisions |
Does the introduction provide sufficient background and include all relevant references? |
Yes |
Thank you for your positive evaluation |
Are all the cited references relevant to the research? |
Yes |
Thank you for your positive evaluation |
Is the research design appropriate? |
Yes |
Thank you for your positive evaluation |
Are the methods adequately described? |
Yes |
Thank you for your positive evaluation |
Are the results clearly presented? |
Can be improved |
Thank you for your feedback. We will place greater emphasis on the presentation of results in our future research. |
Are the conclusions supported by the results? |
Yes |
Thank you for your positive evaluation |
3. Point-by-point response to Comments and Suggestions for Authors |
||
Comments 1: It seems a bit awkward to overuse the first author name of each reference in the introduction or discussion section. It is sufficient to describe only the main content and meaning of the reference.
|
||
Response 1: Thank you for your suggestion. We have revised the introduction and discussion sections to reduce the overuse of first author names. The focus is now placed on summarizing the main content and significance of the references, ensuring a more natural and concise flow of the text. Changes in the text: see Pages 2, 3, 17; line 60, 79, 96, 495-496, 498.
|
||
Comments 2: What is Xc−? Is it different from xCT? Please explain. |
||
|
||
Response 2: Thank you for your question. Xc⁻ refers to the cystine/glutamate antiporter system, which plays a critical role in cellular antioxidant defense by facilitating the uptake of cystine in exchange for glutamate. xCT, on the other hand, is the specific protein subunit (SLC7A11) that forms the functional core of the Xc⁻ system. In short, Xc⁻ is the transporter system, while xCT is the protein component responsible for its activity. Changes in the text: not applicable.
Comments3: In the flow cytometry results for ROS levels, the notation Relative ROS levels is written on the vertical axis, which gives the illusion that the height of the plot represents the ROS level. As far as I understand, the ROS level corresponds to the position on the horizontal axis, so this needs to be distinguished. Also, please provide information on cell counts so that we can know the number of cells corresponding to each plot in the flow cytometry.
Response 3: Thank you for your observation. We agree that the current label on the y-axis may cause confusion. In the revised manuscript, we have moved the label "Relative ROS levels" to the x-axis. Additionally, Figures 3A and 3C represent the relative cell count data from three independent experiments comparing the experimental and control groups. Specific changes can be seen in Figures 3A-D and Figures 6E、G. Changes in the text: see Pages 11,15; Figures 3A-D and Figures 6E、G.
Comments 4: Images of nude mice in tumor removal status do not need to be presented in the main figure.
Response 4: Thank you for your suggestion. We initially included the images of nude mice in the tumor removal status in the main figure to provide a comprehensive and accurate depiction of the experimental process, ensuring that readers could easily understand and trust the results. By including these images in the main figure, our aim was to offer a clearer and more complete description of the experimental setup. As a result, we placed the images of nude mice before and after tumor removal in the main figure. Response 4: Not applicable.
|
- Response to Comments on the Quality of English Language
Point 1: None
Response 1: Not applicable.
- Additional clarifications
We have made every effort to improve the manuscript and have implemented several revisions. These adjustments do not alter the overall content or structure of the paper. We have not listed all the changes here; instead, we have highlighted the revised sections in red text within the manuscript. We sincerely appreciate your hard work and hope that our revisions meet your expectations. Once again, we are truly grateful for your insightful comments and suggestions.
Round 2
Reviewer 2 Report
Comments and Suggestions for Authors
I appreciate the authors careful considerations of my previous comments. Although the authors didn't provide more experimental results to address my comments, the quality of the manuscript has been improved, and I don't have additional comments.